# The Role of Fibroblast Growth Factor 23 in Inflammation and Anemia

**DOI:** 10.3390/ijms20174195

**Published:** 2019-08-27

**Authors:** Brian Czaya, Christian Faul

**Affiliations:** Division of Nephrology, Department of Medicine, The University of Alabama at Birmingham, Birmingham, AL 35294, USA

**Keywords:** fibroblast growth factor 23 (FGF23), fibroblast growth factor receptor (FGFR), klotho, phosphate, chronic kidney disease (CKD), inflammation, anemia, hepcidin, erythropoietin (EPO)

## Abstract

In patients with chronic kidney disease (CKD), adverse outcomes such as systemic inflammation and anemia are contributing pathologies which increase the risks for cardiovascular mortality. Amongst these complications, abnormalities in mineral metabolism and the metabolic milieu are associated with chronic inflammation and iron dysregulation, and fibroblast growth factor 23 (FGF23) is a risk factor in this context. FGF23 is a bone-derived hormone that is essential for regulating vitamin D and phosphate homeostasis. In the early stages of CKD, serum FGF23 levels rise 1000-fold above normal values in an attempt to maintain normal phosphate levels. Despite this compensatory action, clinical CKD studies have demonstrated powerful and dose-dependent associations between FGF23 levels and higher risks for mortality. A prospective pathomechanism coupling elevated serum FGF23 levels with CKD-associated anemia and cardiovascular injury is its strong association with chronic inflammation. In this review, we will examine the current experimental and clinical evidence regarding the role of FGF23 in renal physiology as well as in the pathophysiology of CKD with an emphasis on chronic inflammation and anemia.

## 1. Introduction

Cardiovascular disease is the leading cause of mortality across all stages of chronic kidney disease (CKD), and common occurrences such as fibroblast growth factor 23 (FGF23) excess, chronic inflammation and iron deficiency have been shown as prominent risk factors [1,2,3,4]. Over recent years, the interconnection between these factors has received considerable attention. This review focuses on the latest advances in understanding the complex role of FGF23 in inflammation and anemia, and how these multi-directional relationships present significant insights into the pathomechanisms of CKD.

## 2. The Evolution and Biology of FGF23

Nearly 50 million years ago, life on earth was flourishing due to the re-population from terrestrial creatures, which progressed life beyond their aquatic habitats [5,6,7,8,9]. Throughout this period, an innumerable amount of gene families, including the fibroblast growth factor (FGF) family, expanded in two phases which managed these new extraordinary demands on all terrestrial systems, in comparison to their aquatic ancestors [10,11,12]. FGFs have been identified in both invertebrates and vertebrates. Prior to chordate evolution, the *FGF* gene family expanded from three to six genes via gene duplication, during their first phase of the metazoan linage [13]. During the early emergence and evolution of vertebrates, the *FGF* gene family was subjected to two large-scale gene duplications during their second phase [13]. This resulted in FGFs acquiring more significant roles in biological processes, such as in embryonic development, organogenesis and metabolic homeostasis [13,14,15]. 

The highly conserved FGF family consists of 22 members in mammals and is divided into three classifications: intracellular, paracrine and endocrine FGFs, based on their mechanisms of action [13,16]. Their conserved intrinsic core domain, which spans ~120 amino acids in length (~30–60% amino acid identity among all FGF members) promotes ligand binding to a distinct superfamily of receptor tyrosine kinases, known as fibroblast growth factor receptors (FGFRs) [10,13,17,18,19,20]. Intracellular FGFs are not secreted by their producing cells and act in an FGFR-independent manner, which enables them to mediate intracellular signaling events [13,21,22]. To date, their notable functions are regulating voltage-gated Na^+^ channels and the electrical excitability of neuronal cells [21,23,24,25]. Paracrine FGFs and endocrine FGFs (also called FGF19 subfamily) mediate their biological responses in an FGFR-dependent manner [10,13]. Unlike paracrine FGFs, which bind heparin or heparin sulfate proteoglycans (HPG) as a co-factor to facilitate FGFR activation and function as differentiation factors in development, endocrine FGFs have reduced affinity for HPG due to topological disparities in their heparin-binding region [13,26,27,28]. This permits the FGF19 subfamily to escape extracellular matrices and operate over long distances, functioning as circulating hormones [29,30,31]. As an alternative, endocrine FGFs employ αKlotho or βKlotho as their co-receptor to promote efficient FGF:FGFR binding [31].

FGF23 is a member of the FGF19 subfamily, which utilizes αKlotho to carry out its physiological functions, targeting the kidney to promote phosphate excretion and enabling the suppression of active vitamin D synthesis [13,31,32,33,34,35]. The evolution of this FGF23:αKlotho network is a ramification of vertebrate evolution, where primitive piscine ancestors such as ostracoderms acquired a boney endoskeleton [36]. In 2000, FGF23 was ultimately identified in the ventrolateral thalamic nucleus of mice, and its biological significance rapidly followed when a missense mutation of the *Fgf23* gene in patients with autosomal dominant hypophosphatemic rickets (ADHR) was identified thereafter [37,38]. Under physiological conditions, FGF23 is predominantly produced by osteocytes as a 32 kDa glycoprotein in response to elevations in serum phosphate levels due to dietary loading or serum vitamin D, where its N-terminal region shares homologies with other FGF family members and interacts with FGFRs, whereas its C-terminal portion binds αKlotho [34,39,40,41,42,43]. 

Prior to FGF23 secretion from bone, post-translational modifications allow the peptide to circulate in the bloodstream as a full-length mature form (also called intact FGF23), which possess biological activity, or circulate as proteolytic cleaved fragments (Figure 1) [38,40,44,45]. This proteolytic cleavage event is executed by subtilisin-like pro-protein convertases, such as furin, which occurs at the consensus sequence Arg^176^-X-X-Arg^179^ and is not present in other FGF family members [38,44,46].

As a circulating biologically active peptide, FGF23 is O-glycosylated at several residues, such as Thr^178^ via polypeptide N-acetylgalactosaminyltransferase 3 (GalNT3), which protects FGF23 from proteolytic cleavage and helps support its intact structure [44,47,48,49]. The half-life of this biologically active form of FGF23 is ~45–60 min in humans, whereas in rodents, appears to be much shorter with ~20–30 min in mice and ~5 min in rats [50,51,52]. 

Furthermore, as proteolytic cleaved fragments in circulation, FGF23 is phosphorylated at multiple amino acid residues, such as Ser^180^ via secretory protein kinase family with sequence similarity-20 member C (FAM20C), which impedes O-glycosylation at Thr^178^ by GalNT3 and results in its biologically inactive form by separating the respective FGFR and αKlotho binding regions [53,54]. A firm regulation of these post-translational modification and processing events are essential for maintaining proper homeostatic distributions. Missense mutations in the FGF23 consensus sequence or in GalNT3 and/or FAM20C have been shown to be detrimental to physiologic processes, where interfering with FGF23 cleavage results in either elevated serum intact FGF23 levels that results in hypophosphatemia, or diminished serum intact FGF23 levels which results in hyperphosphatemia due to excessive proteolytic cleavage [38,47,55,56,57,58,59].

Regarding the metabolic rate of FGF23, renal extraction appears to play a critical role in its metabolism, yet only a minor contribution of FGF23 excretion via the kidney may contribute since FGF23 is not detectable in urine under physiological conditions [52]. However, in patients with acute kidney injury, (AKI), FGF23 can be measured following urine analysis, where these elevations correspond with all-cause mortality [60]. Future investigations are needed in this context to help distinguish the origin of urinary FGF23.

## 3. Canonical and Non-Canonical FGF23-Mediated Signaling

The mammalian proteome encodes 58 distinct receptor tyrosine kinases and from this superfamily, four functional *Fgfr* genes (*Fgfr1–4*) have been identified [10,61,62]. Throughout vertebrate evolution, as previously alluded, numerous gene families were subjected to expansions within two distinct phases. The *Fgfr* gene family endured only one expansion in chordate evolution via two large scale gene duplications, which transpired during early vertebrate evolution in the second phase [10]. Human *Fgfr* genes contain eighteen coding exons, which have undifferentiated exon-intron organizational structure and are the principal receptors which mediate the biological effects of FGFs to their respective target tissues [10,19,31]. 

In addition to being ubiquitously expressed throughout the genome, their monomeric structure constitutes three extracellular immunoglobulin (Ig)-like domains, containing the ligand-binding pocket formed by Ig-like domains II and III, a single transmembrane helix and a split intracellular tyrosine kinase domain [19,20,31]. What distinguishes FGF ligand specificities to specific FGFRs and target tissues is the acquisition of alternative mRNA splicing within Ig-like domain III, resulting in IIIb or IIIc isoforms, which increases functional diversity [10,19,20,26]. Furthermore, FGFs require the presence of co-receptors to efficiently bind FGFRs to induce a formation in a 2:2:2 stoichiometry of the FGF:FGFR:co-factor complex [19,20,62,63,64]. Following the binding of FGF ligands prompting lateral contacts between two monomeric receptors, which results in their dimerization, trans-phosphorylation occurs on specific tyrosine residues within their cytoplasmic tail, leading to downstream signaling pathways which converge in the nucleus to induce changes in cellular gene expression [19,20,31,65,66].

FGFR signaling is predominantly transduced by the cytoplasmic effectors, phospholipase Cγ (PLCγ) and FGF receptor substrate 2α (FRS2α) [19,31,61,67]. Subsequent to ligand-induced auto-phosphorylation of FGFRs, PLCγ directly binds to a specific tyrosine residue via its Src homology 2 (SH2) domain within the YLDL (tyrosine-lysine-aspartate-lysine) consensus sequence of the FGFR cytoplasmic tail [68,69,70]. This results in PLCγ activation by the receptor and PLCγ-catalyzed production of diacylglycerol and inositol 1,4,5-triphosphate, which increases cytoplasmic Ca^2+^ levels and leads to the activation of the calcineurin/nuclear factor of activated T-cells (NFAT) pathway [19,65,71,72]. FGFR signaling can alternatively be transduced via activation of FRS2α by FGFR-mediated tyrosine phosphorylation [20,31,67,73]. As opposed to PLCγ, FRS2α is constitutively bound to the juxtamembrane region of the FGFR via its phosphotyrosine-binding (PTB) domain, independent of the receptors activation state [20,74]. FRS2α-mediated signaling results in the activation of Ras/mitogen-activated protein kinase (MAPK) and phosphatidylinositol 3′-kinase (PI3K)/Akt signaling, which the majority of FGFs, such as FGF23, and FGFR isoforms exhibit to mediate their cellular effects [19,61,65,75,76]. 

Canonical FGF23-mediated signaling employs αKlotho as its co-receptor to conduct its physiological activity (Figure 2) [73,77,78,79,80]. Biochemical binding studies have shown that αKlotho increases the binding affinity of FGF23 to FGFRs by ~20-fold [65,81]. Furthermore, the atomic structure for canonical FGF23-mediated signaling revealed that FGF23, αKlotho and Fibroblast growth factor receptor 1c (FGFR1c) form a 1:1:1 monomeric complex, where αKlotho operates as a scaffold protein which tethers the Ig-like domain III of FGFR1c and the C-terminal region of FGF23, implementing FGF23:FGFR1c proximity and conferring the stabilization of the ternary complex [77]. Strikingly, the demand of heparin sulfate is required for the formation of the active signaling complex where FGF23, αKlotho, FGFR1 and heparin sulfate form a 2:2:2:2 stoichiometry to induce complex dimerization, FGFR activation and subsequent activation of MAPK signaling, despite FGF23 having reduced affinity towards heparin sulfate [77]. What defines canonical target tissues for FGF23, is the restriction of αKlotho expression in tissues such as renal tubules and parathyroid gland [75,82,83].

In the kidney, FGF23 binds αKlotho and FGFR1c to promote phosphaturia via the downregulation of the sodium-dependent phosphate co-transporters, NaPi-2a and NaPi-2c, at the apical surface of the proximal tubule [34,84,85]. In addition, FGF23 alters vitamin D metabolism by downregulating the expression of 1-α-hydroxylase (CYP27B1), the enzyme responsible for synthesizing calcitriol from its precursor, and increasing the expression of 24-hydroxylase (CYP24A1), the enzyme which inactivates calcitriol in the renal tubules, thereby reducing circulating levels of active vitamin D [34,39,86]. Likewise, FGF23 binds FGFR1c:αKlotho in the parathyroid to induce the activation of early growth-responsive 1 (Egr-1), which reduces the synthesis and secretion of parathyroid hormone (PTH) [75,87,88,89]. Collectively, FGF23 fosters a negative phosphorus distribution by functioning as a phosphaturic hormone that reduces phosphate absorption from the intestine and increases phosphate excretion by the kidney, thus aiding in the prevention of hyperphosphatemia in the early stages of reduced renal function [73,85].

Prevailing research has elucidated the existence of non-canonical FGF23-mediated signaling, which occurs in an FGFR-dependent, αKlotho-independent manner (Figure 2) [65,90,91,92,93,94,95,96]. In the absence of αKlotho, FGF23 binds FGFR4 with low affinity, leading to PLCγ/calcineurin/NFAT activation due to increasing cytoplasmic Ca^2+^ levels in cultured cells such as cardiac myocytes, hepatocytes and injury-primed renal fibroblasts [90,92,93,94,97,98,99]. These in vitro findings are supported by data originating from various rodent models with elevated serum FGF23 levels, such as administration of recombinant FGF23 in wild-type mice, renal ablation in rats, αKlotho hypomorphic mice and administration of high phosphate or adenine diets in mice, which display adverse outcomes that are associated with CKD such as renal fibrosis, systemic inflammation, anemia and cardiac hypertrophy [90,92,98,100,101,102,103]. These examples demonstrate that αKlotho is not a prerequisite for FGF23 responsiveness and indicates non-canonical tissues, such as the heart and liver, are capable of responding to elevations in serum FGF23 levels. 

Despite these notable pathological conditions, it is feasible for cells that are lacking αKlotho, but are FGF23 responsive, to express alternative co-receptor(s) for FGF23 binding such as cell adhesion proteins of the cadherin and immunoglobulin superfamilies, which might interact with FGFR4 [104,105,106,107,108,109]. To date, such factors have not been identified. In addition to FGFR4, it has also been shown that FGF23 binds FGFR2 on the surface of neutrophils. Following FGF23:FGFR2 complex dimerization and activation, the activation of protein kinase A (PKA) ensues and leads to the inhibition of a small GTPase known as Ras-proximate-1 (Rap1), and the deactivation of β_2_-integrin on the cell surface [95]. As a result, neutrophil arrest on the endothelium and their ability for transendothelial migration is diminished, leading to impaired leukocyte recruitment and host defense [95].

In addition to downstream signaling events resulting from intact FGF23, furin-mediated cleavage results in the separation of the respective FGFR and αKlotho binding domains, generating N-terminal and C-terminal fragments [38,44]. These two fragments in solitary are presumed inactive, which are endorsed by injection studies in mice that demonstrate both fragments lack phosphaturic activity [44]. This outlook has also been disputed, where additional experiments demonstrate C-terminal FGF23 maintains phosphaturic activity, thus insinuating its cellular response is FGFR-independent and perhaps mediated by alternative receptors, or the fragment itself can associate with FGFRs [42,110,111]. Furthermore, a recent study elucidating the crystal structure of the FGF23:FGFR1:αKlotho complex shows that C-terminal FGF23 binds αKlotho between its two extracellular domains, KL1 and KL2 [77]. Moreover, injection studies that employ the C-terminal FGF23 fragment disclose an inhibitory action, which reduces systemic inflammation and anemia in a mouse model of CKD [112]. 

It is plausible to speculate that C-terminal FGF23 competitively blocks the action of intact FGF23, due to binding αKlotho and not activating FGFRs, thus contributing to the inhibition of canonical FGF23-mediated signaling by acting as an antagonist for the FGFR:αKlotho complex. The existence of such a mechanism is supported by an in vitro study which shows that in the presence of C-terminal FGF23, the FGF23-mediated reduction of phosphate uptake by proximal tubule cells can be diminished [113]. Nevertheless, additional studies are needed to elucidate the underlying mechanism.

## 4. FGF23 in Chronic Kidney Disease

The kidneys are part of an inextricable network known as the bone-kidney-intestinal axis [36]. The crucial role of this axis is to regulate mineral metabolism by altering tubular resorption of serum calcium and phosphate, with the assistance of key modulators FGF23, PTH and 1-25-dihydroxyvitamin D [85,114]. In an average Western diet, individuals ingest ~1200 mg/day of phosphate, yet a net weight of ~900 mg/day is absorbed into circulation [115,116,117]. The amount of phosphate absorption depends on its exogenous source and bioavailability [118,119]. Phosphate exists in two forms, organic and inorganic phosphate [85,119]. 

Organic phosphate is predominantly found in foods that are rich in protein (non-phytates) such as meat, fish and dairy products, which have a bioavailability of 40–80% [85,119,120,121,122]. By contrast, organic phosphate can also be obtained from plant-derived foods (phytates) such as cereal and nuts, but they have a bioavailability less than 40% [85,123,124,125]. This dissimilarity in phosphate absorption is attributed to monogastric mammals lacking the enzyme phytase, which is acquired to liberate phosphate [85,118,119,126]. To increase dietary absorption, phosphate additives are utilized to increase shelf life, alter texture and increase the flavor of food products [127,128,129,130]. These inorganic phosphate sources are passively absorbed in the intestine and have a bioavailability of nearly 100% [36,85,131,132]. With a wide range of foods containing additives such as polyphosphates and pyrophosphates, the relation of nephrotoxicity to phosphate toxicity (phosphotoxicity) has been the focus of many studies [36,119,133,134,135,136,137,138]. Clinical studies have shown that a single dose of 11.5 g of phosphate can accelerate de novo CKD over the course of several months [36].

To prevent phosphotoxicity, FGF23 levels rise and fall in correlation with the amount of dietary phosphate absorbed [139,140]. When an individual absorbs phosphate from foods that are high in its bioavailability, the bone upregulates FGF23 production to induce greater urinary excretion of phosphate and minimizing the efficiency of intestinal phosphate absorption by reducing serum 1-25-dihydroxyvitamin D levels [139]. In contrast, when an individual absorbs phosphate from foods that are low in its bioavailability, renal phosphate reabsorption is enhanced, as well as its absorption efficiency from the intestine by 1-25-dihydroxyvitamin D [139]. When renal damage occurs and kidney function declines, the derangement of bone and mineral homeostasis is inescapable. This event is termed chronic kidney disease-mineral bone disorder (CKD-MDB), which refers to renal dysfunction and altered levels of calcium, phosphate, PTH, 1-25-dihydroxyvitamin D and FGF23 [140,141].

During the early stages of CKD, the expression of the renal FGF23 co-receptor, αKlotho, decreases in response to kidney damage and progressively declines along with the loss of functional nephrons, promoting partial resistance to FGF23′s physiological actions [139,142,143,144]. As a compensatory mechanism, FGF23 levels rise 1000-fold above normal values in an attempt to maintain a neutral phosphate distribution [139,145,146]. This compensatory increase in FGF23 promotes the suppression of 1-25-dihydroxyvitamin D production, which in turn, promotes the elevation of PTH causing secondary hyperparathyroidism [147]. Throughout these hormonal alterations, serum phosphate levels gradually increase and by end-stage renal disease (ESRD), ultimately culminate in overt hyperphosphatemia due to renal resistance to FGF23′s actions on impaired kidneys [115,139,147]. 

These metabolic disturbances primarily contribute to associated pathologies that are observed in CKD, such as immune dysfunction, systemic inflammation, anemia, vascular calcification, skeletal muscle atrophy and cardiac hypertrophy, resulting in premature death [148,149,150,151,152,153,154]. Although FGF23 acts in a compensatory manner towards these elevating phosphate levels, clinical CKD studies have demonstrated powerful and dose-dependent associations between elevations in serum levels of FGF23 and CKD-associated pathologies, such as systemic inflammation, anemia and cardiovascular mortality, which is the leading cause of death across all stages of CKD [2,155,156,157,158]. FGF23 has also been shown to directly promote immune dysfunction, systemic inflammation and cardiac hypertrophy [90,92,95,98].

### 4.1. Inflammation and Iron Deficiency, Two Novel Determinants of FGF23 Production 

Systemic immune activation and alterations in iron trafficking are interlinked in CKD, as the inflammatory milieu contributes to the pathology of functional iron deficiency [159,160,161]. Pro-inflammatory cytokines such as interleukin-1β (IL-1β), interleukin-6 (IL-6) and Oncostatin M (OSM) promote the upregulation of hepcidin in the liver [162,163,164,165,166,167,168,169]. As a consequence, hepcidin promotes iron dysregulation by inhibiting gastrointestinal iron absorption and blocking the release of recycled iron from the reticulo-endothelial system into plasma [159,170,171,172,173]. Experimental studies suggest both inflammation and iron deficiency are novel regulators of systemic FGF23 synthesis and secretion [161,174].

Many clinical studies have reported associations between FGF23 and inflammatory markers in disease states [175,176,177,178]. Furthermore, it has been shown that acute and chronic inflammation directly promote the production of FGF23. Utilizing two distinct animal models of acute inflammation, David et al. have revealed that a single intraperitoneal injection of either heat-killed Brucella abortus or IL-1β in wild-type mice results in a 10-fold increase of *Fgf23* mRNA levels in bone and serum levels of C-terminal FGF23 protein, yet serum intact FGF23 levels were unaltered [179]. Moreover, the induction of functional iron deficiency without superimposed inflammation was produced by administration of exogenous hepcidin into wild-type mice, where bone expression and C-terminal FGF23 levels were significantly elevated as a result of the stabilization of hypoxia-inducible factor 1α (HIF1α) and its subsequent translocation and binding to hypoxia response elements on the Fgf23 promoter [179]. This dissimilarity between serum C-terminal and intact FGF23 levels were shown to be attributed to concomitant increases in coupling FGF23 production to its cleavage [161,179].

Equally, IL-6 has been shown to be a novel regulator of FGF23. In the state of acute and chronic inflammation, IL-6 elevates the synthesis and secretion of FGF23 from bone by inducing the phosphorylation of signal transducer and activator of transcription 3 (STAT3) through soluble IL-6 Receptor (sIL-6R)-mediated trans-signaling, prompting STAT3 to bind its response element on the Fgf23 promoter [180]. Utilizing animal models of CKD and AKI where serum IL-6, sIL-6R and FGF23 levels were significantly elevated, Durlacher-Betzer et al. showed that mice with global deletion of IL-6 displayed reduced levels of serum FGF23 in comparison to control mice, thus exhibiting IL-6 as a potent mediator of FGF23 [180]. 

Additional mediators of the inflammatory milieu, such as tumor necrosis factor-α (TNF-α) and OSM, have also been shown to stimulate FGF23 synthesis and secretion. In cultured bone cells and following a recent study utilizing animal models of renal and extrarenal inflammation, Glosse et al. showed that increasing concentrations of TNF-α elevates the production of FGF23 in a dose dependent-manner in UMR106 cells, while Egli-Spichtig et al. revealed the contribution of TNF-α stimulation to the renal production of FGF23, respectively [181,182]. In addition to these reservoirs, Richter et al. displayed that increasing concentrations of OSM coincided with increased FGF23 production in cardiac myocytes, thus exhibiting the heart as a novel source of FGF23 production [183]. In tandem with these findings, a genome-wide investigation into FGF23 has revealed a distal enhancer upstream of the FGF23 transcriptional start site that contributes to these elevated FGF23 levels, following these various inflammatory stimuli [184]. Deletion of this putative 16 kb enhancer region in the mouse genome significantly blunts elevated FGF23 levels in circulation [184].

In contrast to functional iron deficiency and its mediators regulating FGF23 production, absolute iron deficiency has been shown to regulate FGF23, along with clinical studies demonstrating an association between FGF23 and erythropoiesis [185,186,187,188,189]. The influence of blood loss with respect to increased FGF23 production became apparent when Rabadi et al. displayed that acute loss of 10% total blood volume in C57BL/6 mice lead to a 20-fold increase in FGF23 mRNA expression in bone marrow, accompanied by acute elevations of serum C-terminal FGF23 and erythropoietin (EPO) levels after 6 h [190]. Furthermore, Hanudel et al. have demonstrated that beta-thalassemia intermedia mice, which exhibit chronically high endogenous EPO levels, and wild-type mice with and without CKD, following a single intraperitoneal injection of exogenous EPO, display significant elevations in FGF23 mRNA levels in bone marrow and serum C-terminal FGF23 levels as opposed to serum intact FGF23 levels [191]. These results are further supported by studies in kidney transplant recipients and patients on dialysis [191]. A plausible mechanism has been implied by Toro et al., where blockade of the EPO receptor prevents the induction of FGF23 synthesis and secretion in primary bone marrow cells isolated from tibias of male C57BL/6 mice [192]. In addition, mice subjected to septic AKI induced by cecal ligation and puncture were associated with elevated EPO and FGF23 levels in the circulation, thus suggesting EPO may also play a casual role in FGF23 induction in AKI [192]. 

Collectively, these data suggest inflammation and iron deficiency are both potent stimuli of FGF23 production. Additional studies are required to investigate the disarranged processing events of FGF23 following conditions of acute and chronic inflammation, absolute iron deficiency and elevated EPO levels. C-terminal and intact FGF23 levels are elevated at dissimilar magnitudes, where during acute settings more biologically inactive FGF23 enters the circulation, yet during chronic settings more biologically active FGF23 exists [161]. Altogether, the crosstalk between inflammation, iron deficiency and FGF23 excess is important for clinical practice, since all three factors are associated with higher risks for mortality.

### 4.2. The Role of FGF23 in Chronic Inflammation

With alterations in phosphate homeostasis, it has been recognized that widespread tissue injury and accelerated CKD progression can be attributed to the presence of chronic inflammation [193,194,195,196]. Clinical studies in CKD, along with other disease populations, have reported positive associations between ascending FGF23 quartiles and elevations in pro-inflammatory cytokines such as IL-6, TNF-α and C-reactive protein (CRP) [157,175,178]. Many studies focus on inflammatory stimuli driving FGF23 production but the existence of non-canonical FGF23-mediated signaling prompts the question if there is a vicious cycle that promotes uncontrolled production of FGF23 together with pro-inflammatory cytokines, overall contributing to adverse outcomes that are associated with CKD (Figure 3).

Indeed, Singh et al. unveiled by the use of in vitro experiments and various animal models of FGF23 excess, that FGF23 can directly target the liver to enhance the inflammatory milieu [92]. Hepatocytes have the highest expression levels of FGFR4 in mammals and lack αKlotho [15,197,198,199,200]. In settings of uncontrolled serum FGF23 levels, FGF23 binds FGFR4 on the surface of hepatocytes to activate PLCγ/calcineurin/NFAT signaling, resulting in increased synthesis and secretion of IL-6 and CRP [92]. Following pharmacological inhibition or deletion of FGFR4, these inflammatory markers and their hepatic production are ameliorated [92]. Considering these results, to date, no studies have reported harmful effects of FGF23 on hepatocytes. This is consistent with clinical observations that the presence of CKD per se does not promote liver injury, despite elevations in serum FGF23 levels [65].

Moreover, FGF23 can also stimulate the production of inflammatory cytokines from other known reservoirs. In patients with chronic obstructive pulmonary disease (COPD) and cystic fibrosis (CF), serum FGF23 levels are significantly elevated [201,202,203,204]. For the first time, Krick et al. have shown that with the exposure of cigarette smoke, which is accountable for a large number of COPD cases, αKlotho expression is downregulated but FGFR4 expression in airway epithelial cells is increased [203]. These cellular alterations allow FGF23 to act on bronchial epithelial cells, which results in the activation of FGFR4/PLCγ/calcineurin/NFAT signaling and subsequent upregulation of IL-1β expression in COPD patients [203]. In addition to COPD, Krick et al. have shown that FGF23 targets CF-human bronchial epithelial cells to increase the secretion of IL-8, which is a key cytokine that contributes to chronic inflammation in CF patients [204,205]. Alongside these novel relationships between FGF23 and consequential interleukins, Han et al. further expanded the role of FGF23 in inflammation by revealing FGF23 targets peritoneal macrophages to upregulate TNF-α production [206].

Apart from these studies, the use of genome wide-analysis has indicated that many pro-inflammatory genes are regulated by FGF23 [184]. Since FGF23-induced cytokine production is primarily attributed to NFAT activation and NFAT induces various cytokine genes such as TNF-α, IL-2, IL-4 and IL-6 in distinct cell types such as T-cells and mast cells, the activation of non-canonical FGFR4-dependent signaling in other cell types and tissues, in conjunction with contributing to systemic elevations of various inflammatory cytokines, is plausible [207,208].

Collectively, these data suggest FGFR4 blockade might be effective in reducing systemic inflammation in patients. It is possible that FGF23-induced cytokine production is not initially pathological, but more so representative of a beneficial effect which transitions into a detrimental consequence due to the prolonged duration of exposure to increased circulating FGF23, rather than the degree of FGF23 elevations per se. Apart from this postulation, it is important to indicate other factors besides circulating FGF23, such as elevated serum phosphate levels, might be culpable for elevations in inflammatory cytokines and perhaps bestows inflammatory tissue damage [209,210]. Therefore, further research is required to determine the direct effects of phosphate on specific tissues, such as the heart and liver. In tandem, investigations whether physiologic concentrations or merely high concentrations of FGF23, as perceived in advanced CKD, stimulate the hepatic production of cytokines and whether these cytokines have physiologic or pathologic functions are needed to elucidate the potential rational for the interconnection between FGF23 and inflammatory milieu. However, as only some animal models examined with elevated FGF23 developed renal injury or hyperphosphatemia, while all animal models displayed increased serum concentrations of cytokines, it is feasible FGF23 acts as a major driver of inflammation in CKD [65].

## 5. Chronic Inflammation, a Silent Culprit of Chronic Kidney Disease

Currently, 40 million adults in the United States suffer from CKD, which corresponds to ~10–15% of the adult population and accounts for 24% of the annual Medicare budget [211,212]. With these expenses, clinical outcomes for patients have not been met with positive results, and alternatively have been confronted with a 30% increase in the prevalence of CKD over the past decade, with increasing quantities of ESRD patients requiring dialysis or transplantation to remain alive [213,214]. CKD is defined as abnormalities of kidney structure and function for over 3 months, along with deleterious implications of the physical state of the patient [215,216,217,218]. The basis of CKD can be evaluated by the progressive loss of renal function, which leads to a reduction in the glomerular filtration rate (GFR) that can be calculated using the Cockcroft–Gault equation, and an increase in serum albumin levels which can be detected via dipstick urinalysis [217]. The progressive nature of this disease is further promoted by the compensation of remaining nephrons in response to hyperfiltration of the kidney, which compounds additional renal injury [219,220,221]. The diagnosis for CKD remains muted, since it is rare in its initial stages and often progressive. Once a patient is diagnosed, it is unfortunately irreversible but clinical treatments attempt to manage this disease to prevent further progression to ESRD [222].

CKD is a state of systemic inflammation, characterized by low to moderate levels of circulating inflammatory markers which have been acknowledged since the late 1990’s [196,223,224,225]. The escalating consequences of chronic inflammation have reshaped previous perceptions and is now identified as a well-established risk factor of morbidity and mortality in pediatric and adult CKD [225,226]. Under physiologic limitations, the inflammatory response fosters beneficial outcomes such as tissue repair [227,228,229]. If a reduction in the renal clearance of cytokines transpire, a chronic inflammatory state emerges with systemic consequences. These circumstances then foster a catalytic environment for pathologic complications in CKD such as muscle wasting, vascular calcification, anemia and cardiac remodeling [196,230,231]. 

Multiple factors contribute to the etiology of the inflammatory state in CKD. One such factor is oxidative stress, which occurs when there is a disproportion of the production to degradation ratio in reactive oxygen species (ROS) generating superoxide anions (O_2_^-^), hydrogen peroxide and hydroxide radicals [232,233,234]. Uremic toxicity, which is inseparably linked to CKD, manufactures oxidative stress by elevating the production of ROS and impairing antioxidant capacity, influencing a pro-oxidant state [196,235,236,237]. This enhanced ROS generation then activates immune cells, such as monocytes and macrophages [196,238]. Consequently, inflammatory cells release reactive substances such as O_2_^-^ at the site of inflammation and injury, which promotes the activation of pro-inflammatory gene programs and further ROS production [196]. Thus, oxidative stress and inflammation are synergistically linked in CKD and this vicious cycle contributes to tissue injury and dysfunction via activation of redox-sensitive transcription factors and signaling pathways which promote necrosis, apoptosis and inflammation [196].

Uremic toxins such as indoxyl sulphate and *p*-cresol also promote intestinal dysbiosis, a further factor which contributes to the etiology of the inflammation in CKD [239,240]. Intestinal dysbiosis triggers intestinal bacteria to translocate into the bloodstream, which permits activation of the immune system and prompts persistent systemic inflammation, contributing to the inflammatory status of the patient [196,241]. Additional factors such as metabolic acidosis and frequent infections also play critical roles in the etiology of inflammation in CKD and accelerate its progression [196]. Genetic studies have also uncovered genetic and epigenetic factors that influence the severity of the inflammatory milieu, in which various approaches such as modifying dietary requirements and dialysis optimization of the patient aim to alleviate the underlying inflammatory status [196,225].

This dynamic inflammatory response requires the attraction of the immune system to the site of injury via production of chemokines and cytokines produced by tissue resident immune cells and apoptotic tissue [242,243,244,245]. Pro-inflammatory mediators synthesized by these immune cells at the site of injury, such as TNF-α, IL-1β and IL-6, further aid the expansion of the inflammatory response to a systemic level by targeting various organs in an endocrine-manner, such as the liver, to promote the production of acute phase proteins to mediate downstream effects [246,247,248]. The result is the initiation of the acute phase response, which is a systemic reaction to tissue disturbances with the objective to alleviate underlying inflammation [248]. The predominant cell type involved in the acute phase response are hepatocytes, which produce type I and type II acute phase proteins such as CRP and hepcidin [248,249], respectively. Other cell types such as macrophages and their distinct subsets also regulate the balance of tissue inflammation and repair in response to the local microenvironment [244]. Together, the micro- and macro-environment facilitate proper polarization of immune cells, which in turn contributes to beneficial effects [244]. Although elevated levels of circulating cytokines help rectify initial tissue damage and influence beneficial results, they have been implemented in accelerating adverse outcomes [225]. Inflammatory mediators such as TNF-α, IL-6 and IL-1β, are significantly higher in CKD patients and are among the strong predictors of poor clinical outcomes [194,250]. Furthermore, they exhibit an inverse correlation with renal function, where a mild to moderate reduction in GFR has been implied as a determining factor [194]. 

TNF-α is a polypeptide cytokine that promotes anti-tumor and immune responses [251]. Experimental evidence has shown that TNF-α directly contributes to muscle wasting and vascular calcification by activating TNF-α receptor type 1, enhancing mitochondrial ROS production and activating nuclear factor-κB (NFκB) signaling, which is a transcription factor and key regulator of a wide array of pro-inflammatory gene programs, such as IL-6 and IL-1β [252,253,254,255]. Nevertheless, additional experimental studies also suggest TNF-α activates cAMP (cyclic adenosine monophosphate) and Wnt-mediated pathways to contribute to vascular calcification [256,257].

Apart from TNF-α, IL-6 is a cytokine that exhibits pleotropic activity and plays a crucial role in processes such as hematopoiesis and the immune response [258,259,260]. Experimental studies have shown, excluding its contribution to host defenses through the regulation of acute phase proteins, IL-6 directly contributes to vascular calcification in vascular smooth muscle cells by promoting osteogenic transition and indirectly by reducing fetuin-A expression in the liver and αKlotho expression in the kidney [261]. Furthermore, IL-6 has been shown to directly contribute to adverse outcomes in CKD such as anemia by upregulating hepcidin production in hepatocytes and inducing cardiac remodeling in myocytes by activating the gp130/IL-6R complex, resulting in phosphorylation and activation of STAT3 [162,262,263,264]. Equally, IL-1β is a potent mediator of the inflammatory response and is crucial for host-defense responses to infection and injury, such as leukocytosis and fever [265,266]. It is secreted by various cell types following the activation of the NLRP3 inflammasome and has been shown to contribute to CKD-associated anemia by upregulating hepcidin production in hepatocytes [163,266]. 

Altogether, these pro-inflammatory cytokines accelerate and contribute to adverse outcomes in CKD, despite being crucial mediators of host-defense responses. The pathophysiology of inflammation in CKD is not identical between patients, but a persistent low-grade inflammatory status has been established as a hallmark feature of CKD. Understanding the role of this hallmark is crucial for the development of future therapeutic interventions, which could reveal new signal transduction pathways that accelerate and contribute to adverse outcomes, particularly the specific etiology of inflammation. Furthermore, the identification of novel biomarkers for inflammation in CKD could further support an earlier diagnosis.

## 6. Mechanisms Underlying Iron Dysregulation in Chronic Kidney Disease

Iron is essential for life and is utilized in countless metabolic processes virtually in all species [267]. In humans, 75% of the total iron content can be attributed to erythrocytes with a modest portion stored in organs such as the liver [268,269,270]. Unlike phosphate, iron is predominantly obtained within the mammalian body from macrophages via phagocytosis of senescent erythrocytes in the reticuloendothelial system [270,271,272]. Erythrocytes undergo alterations to distinct surface markers, such as the redistribution of phosphatidylserine from the inner leaflet to the outer leaflet of the cell membrane, which allow these erythrocytes to be recognized for erythrophagocytosis at the end of their lifespan [273,274,275,276]. Once digested by macrophages, iron is then recovered by the actions of heme oxygenase [270,273]. This process efficiently recycles and exports iron back into circulation for hemoglobin synthesis [272]. Although iron metabolism and its bioavailability are tightly regulated at both the cellular and systemic level, disruptions in iron homeostasis for CKD patients have been recognized as a common complication, yet this disorder for patients with early CKD often goes undiagnosed or untreated. 

Anemia is defined as hemoglobin levels that are less than 13 g/dL in men and 12 g/dL in women which are older than fifteen years of age [277]. The prevalence of this condition increases from 5% in patients with an estimated GFR (eGFR) greater than 60 mL/min/1.73 m^2^ to 40% in those with an eGFR less than 30 mL/min/1.73 m^2^, affecting nearly all patients with ESRD [278,279]. Together with the progressive loss of renal function, anemia increases the likelihood of muscle myopathy, cognitive impairment, cardiovascular disease and frequent hospitalization for patients [280,281,282]. Moreover, patients may be diagnosed with various forms of iron deficiency which occur in renal anemia. Of these, absolute iron deficiency and functional iron deficiency play a central role [278,283].

### 6.1. Absolute Iron Deficiency

The pathophysiology of absolute iron deficiency (AID) results from a reduction in the body’s total iron content, where the rate of dietary iron absorption cannot compensate for the increased demand for iron or blood loss [283]. What characterizes AID is transferrin saturation (TSAT) levels that are less than 20% and serum ferritin levels less than 100 ng/mL, which depicts inadequate iron stores [283]. Several mechanisms contribute to this condition such as EPO deficiency, nutritional insufficiencies, retention of metabolites which impair erythropoiesis and increased blood loss [284,285,286,287,288,289,290,291,292,293].

EPO is a kidney-derived hormone which stimulates the production of erythrocytes by regulating the differentiation of hematopoietic progenitor cells, such as pro-erythroblasts, in bone marrow [294]. Experimental studies have shown EPO is synthesized and secreted by fibroblast-like cells in the peri-tubular interstitum of the cortex and outer medulla [295,296]. In the presence of kidney damage, peri-tubular fibroblasts undergo trans-differentiation into myofibroblasts [294]. As a result, EPO deficiency transpires due to the inability of myofibroblasts to secrete EPO [294]. In addition, studies have shown bone marrow cultures incubated with uremic sera generated by metabolite retention impair cell growth of both erythroid burst-forming units and erythroid colony forming units [290]. Furthermore, the increase in blood loss can result from uremic-associated platelet dysfunction, common procedures following hospitalizations such as phlebotomy, hemodialysis or surgical procedures that result in gastrointestinal bleeding [297]. 

As AID poses a major risk factor for CKD-associated pathologies, traditional treatments such as erythropoietin stimulating agents, intravenous iron, oral iron supplementation and blood transfusions have been widely employed to alleviate the condition. Clinically, the etiology of AID should be addressed early since the improper diagnosis between AID and functional iron deficiency could be detrimental. An example of this issue is whether a CKD patient with TSAT levels below 20% and serum ferritin levels above 500 ng/mL would benefit from iron supplementation [171]. Patients that are not properly diagnosed have high probabilities of suffering from systemic iron toxicity, since iron supplementation therapy may result in hemochromatosis due to chronic infections and inflammation promoting cellular iron retention [171].

### 6.2. Functional Iron Deficiency

Likewise, CKD patients have high likelihoods of developing functional iron deficiency (FID) [278]. This complex disorder is characterized by TSAT levels that are less than 20%, due to bone marrow extracting iron from the circulation faster than the rate at which iron is released from stores, and in combination with serum ferritin levels that are normal or higher than 200 ng/mL, which depicts adequate or high iron stores [283,298]. The pathophysiology of FID is a consequence of systemic inflammation causing insufficient iron mobilization [273]. Pro-inflammatory cytokines induce the synthesis and secretion of the iron regulatory hormone hepcidin, which is predominately produced by the liver [299]. In normal physiology, this regulatory event plays a critical role in host defense by withholding iron from invading pathogens to modulate their growth and pathogenicity [300,301]. However, in CKD, hepcidin excess contributes to the restriction of iron for erythropoiesis and anemia [171,273]. Hepcidin regulates systemic iron levels and exerts its iron-regulatory function by binding the sole known transmembrane iron exporter, ferroportin (FPN) [302]. 

This hepcidin-FPN interaction triggers the internalization of FPN from the surface of cells such as hepatocytes, macrophages and enterocytes, which leads to FPN degradation by the lysosome [159,273]. As for systemic ramifications of FPN degradation, gastrointestinal absorption of iron and the exportation of recycled iron from macrophages within the reticuloendothelial system to the bloodstream is blocked and serum iron levels significantly reduce [270]. Clinically, the etiology of FID has received considerable attention. Pharmaceutical approaches targeting this hepcidin-FPN axis, in regard to correcting inappropriate iron retention and improving systemic iron mobilization, has been the focus of numerous studies [303]. Iron redistribution therapies such as hepcidin neutralizing antibodies, anti-chalins and aptamers have been proven useful and successful in phase I clinical trials to reduce circulating levels of hepcidin [303,304]. 

Furthermore, approaches tackling hepcidin-induced internalization of FPN have been subjected to examination. Agents such as FPN stabilizing antibodies have been utilized to block FPN degradation in phase II clinical trials [303,304]. In addition, bone morphogenic protein (BMP) signaling inhibitors specifically for BMP2 and BMP6, along with treatments impacting the severity of inflammation have been employed to inhibit hepcidin synthesis [303,304]. Although these diverse approaches appear to be beneficial, additional investigations into alternative targets are urgently needed to support a balanced iron homeostasis in patients with CKD. 

## 7. Direct and Indirect Actions of FGF23 in Anemia

Anemia in CKD is a multifactorial process and with traditional origins, such as impaired erythropoiesis and iron deficiency, studies have devoted considerable attention towards unraveling the contribution of FGF23 to renal anemia [185,281,305,306]. Clinical investigations have demonstrated significant associations between increased serum FGF23 levels, the longitudinal decline in hemoglobin levels and the overall risk incidence of anemia [158,307]. Furthermore, recent studies have displayed an inverse correlation with transferrin saturation levels, serum iron levels and erythropoiesis [112,308,309,310]. The pleiotropic actions of FGF23, along with its functional impact in renal anemia, have been ascribed to direct and indirect mechanisms (Figure 4).

The formation of bone, also known as osteogenesis, is essential for erythrocyte and platelet production in bone marrow [311]. Experimental evidence has shown that postnatal depletion of osteoblasts results in bone loss and detrimental outcomes for hematopoiesis, which is evident by severe reduction in B-lymphocytes, erythrocytes and hematopoietic stem cells (HSC) [312,313]. This reduction in bone mineralization also leads to inadequate localization of HSC to the endosteal niche [314]. Furthermore, mice possessing global FGF23 deletion display severe bone abnormalities. To answer the question if FGF23 plays a role in erythropoiesis, Coe et al. utilized mice with global deletion of FGF23 and tools such as flow cytometric analysis to uncover the direct actions of FGF23 [310]. With this genetic inactivation of FGF23, profound heightened frequencies in HSC and erythroid progenitor cells in bone marrow and peripheral blood were observed. In addition, serum EPO levels were significantly higher than wild-type controls. Critically, a single intraperitoneal injection of exogenous FGF23 in wild-type mice resulted in contrasting effects. Moreover, utilizing an FGF23 blocking peptide to inhibit FGF23 signaling and 5/6 nephrectomy (Nx) mice, which develop anemia 8 weeks post-surgery, Agoro et al. showed a single intraperitoneal injection of FGF23 blocking peptide was sufficient to rescue renal anemia in this Nx model [112]. These results were accompanied by the lower frequency of erythroid cell apoptosis. Collectively, these data suggest elevated serum FGF23 levels negatively regulate erythropoiesis and are a negative mediator of HSC’s differentiation into the erythroid linage.

As for the indirect actions of FGF23, the interconnection between FGF23 and pro-inflammatory cytokines, such as in CKD, provides further insight on how FGF23 indirectly contributes to anemia. As previous alluded, excess serum FGF23 levels directly stimulate the synthesis and secretion of various pro-inflammatory cytokines such as IL-6 and IL-1β, which triggers the formation of a vicious cycle that promotes the uncontrolled production of both factors. As a consequence of this enhanced inflammatory milieu, cytokines upregulate the production of hepcidin in the liver. These pathologic events foster excessive serum hepcidin levels, which in turn, promotes functional iron deficiency. 

## 8. Conclusions

In summary, abnormalities in mineral metabolism and the metabolic milieu are associated with chronic inflammation and anemia, and FGF23 is a risk factor in this context. Recent investigations have shown that inflammation and alterations in both EPO and iron status, are potent inducers of FGF23 production. As for the multi-directional relationship between factors, FGF23 exerts pleiotropic actions as a positive mediator of the inflammatory response and a negative regulator of erythropoiesis. Clinical and laboratory findings have revealed novel insights into the pathomechanisms of CKD and the establishment of a feed-forward cycle promoting the dysregulation between factors has led to the identification of novel therapeutic opportunities, such as targeting non-canonical FGF23-mediated signaling.

However, further studies are necessary to investigate the disarranged processing events of FGF23 following conditions of inflammation and anemia, where C-terminal and intact FGF23 levels are elevated in the circulation at dissimilar magnitudes. Moreover, these studies should aim to answer a key question, whether N-terminal and C-terminal cleaved fragments possess biological activity and if so, whether these activities are alike or disparate to the functions of intact FGF23. 

## Figures and Tables

**Figure 1 ijms-20-04195-f001:**
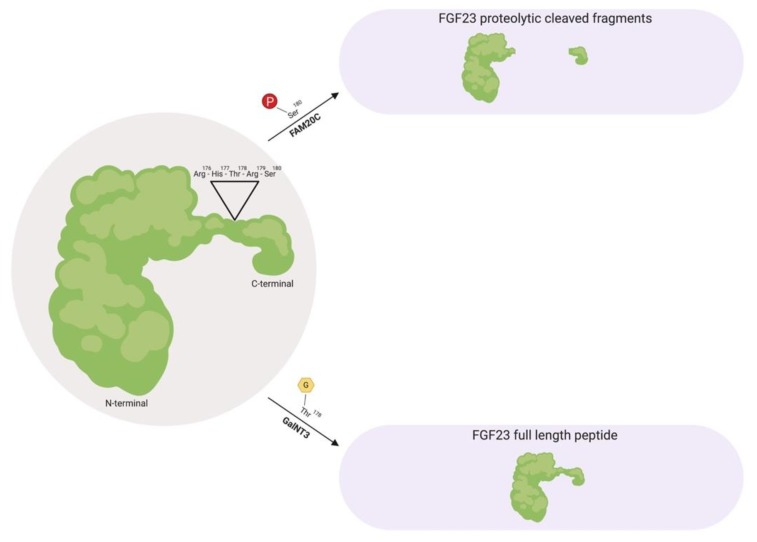
Fibroblast growth factor 23 (FGF23) regulation by post-translational modification. As a full-length biologically active peptide in circulation, FGF23 is O-glycosylated by GalNT3 at several residues such as Thr^178^, which protects FGF23 from proteolytic cleavage by pro-protein convertases such as furin. Vice versa, as proteolytic cleaved fragments in circulation, FAM20C phosphorylates FGF23 at multiple amino acids, such as Ser^180^. This phosphorylation event impedes O-glycosylation by GalNT3 and allows furin to recognize its Arg^176^-His^177^-Thr^178^-Arg^179^-Ser^180^ consensus sequence in FGF23, thus leading to FGF23 cleavage and separation of the N-terminal and C-terminal fragments. GalNT3, polypeptide N-acetylgalactosaminyltransferase 3; FAM20C, secretory protein kinase family with sequence similarity-20 member C; Arg, arginine; His, histidine; Thr, threonine; Ser, serine.

**Figure 2 ijms-20-04195-f002:**
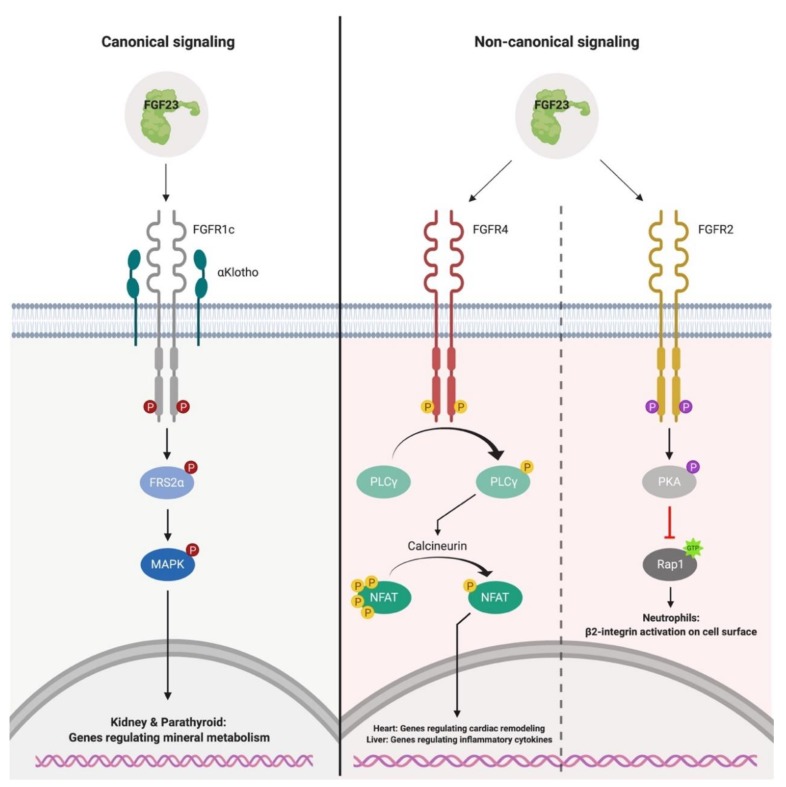
Canonical and non-canonical signaling of FGF23. Canonical FGF23-mediated signaling occurs in an αKlotho-dependent manner in the kidney and parathyroid, leading to the phosphorylation of FRS2α and MAPK. Activation of this signaling pathway results in the downregulation of the sodium-dependent phosphate co-transporters, NaPi-2A and NaPi-2c, on the apical surface of the proximal tubule. In the parathyroid, canonical signaling results in Egr-1 activation and suppression of PTH production. Non-canonical FGF23-mediated signaling occurs in an αKlotho-independent manner in cell types such as myocytes, hepatocytes and neutrophils. In myocytes and hepatocytes, FGF23 binds FGFR4 to activate the PLCγ/calcineurin/NFAT pathway, leading to the induction of gene programs involved in cardiac remodeling and production of pro-inflammatory cytokines. In neutrophils, FGF23 binds FGFR2 which leads to the activation of PKA and subsequent inhibition of the GTPase, Rap1. As a result, β_2_-integrin deactivation occurs on the cell surface and leads to impaired leukocyte recruitment and host defense. FRS2α, FGF receptor substrate 2α; MAPK, Ras/mitogen-activated protein kinase; Egr-1, early growth-responsive 1; PTH, parathyroid hormone; PLCγ; phospholipase Cγ; NFAT, nuclear factor of activated T-cells; PKA, protein kinase A; Rap1, Ras-proximate-1.

**Figure 3 ijms-20-04195-f003:**
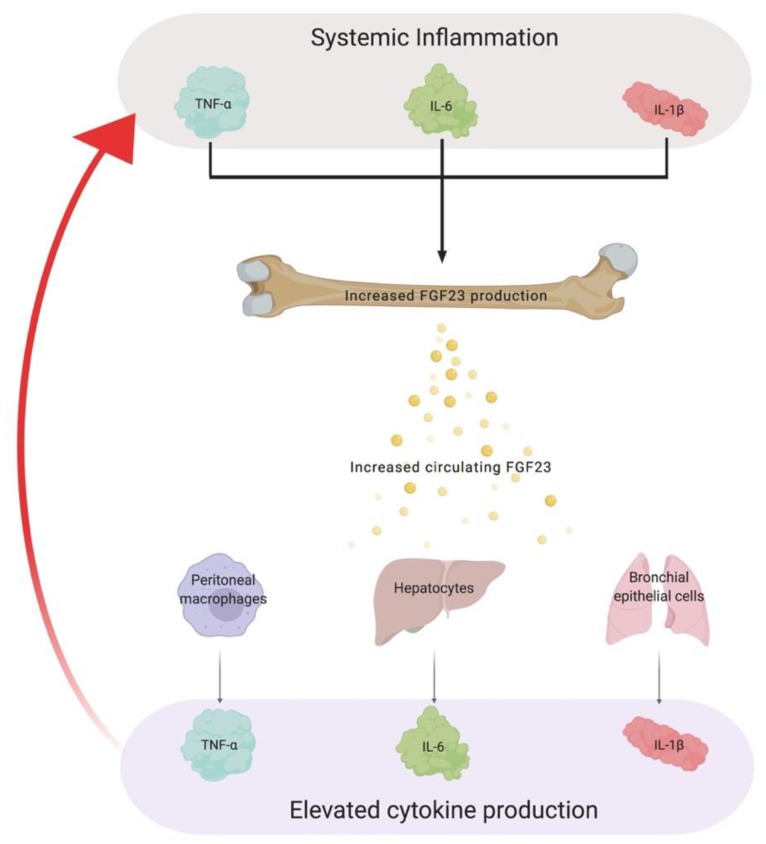
A vicious cycle connecting FGF23 and chronic inflammation. Pro-inflammatory cytokines such as TNF-α, IL-6 and IL-1β are potent inducers of FGF23 production. As a consequence of these factors promoting the uncontrolled production of FGF23, excess levels of FGF23 in circulation elevate the production of various cytokines in diverse tissues, which in turn, amplifies systemic inflammation. The establishment of this vicious cycle contributes to widespread tissue injury and the acceleration of CKD. TNF-α, tumor necrosis factor-α; IL-6, interleukin-6; IL-1β, interleukin-1β; CKD, chronic kidney disease.

**Figure 4 ijms-20-04195-f004:**
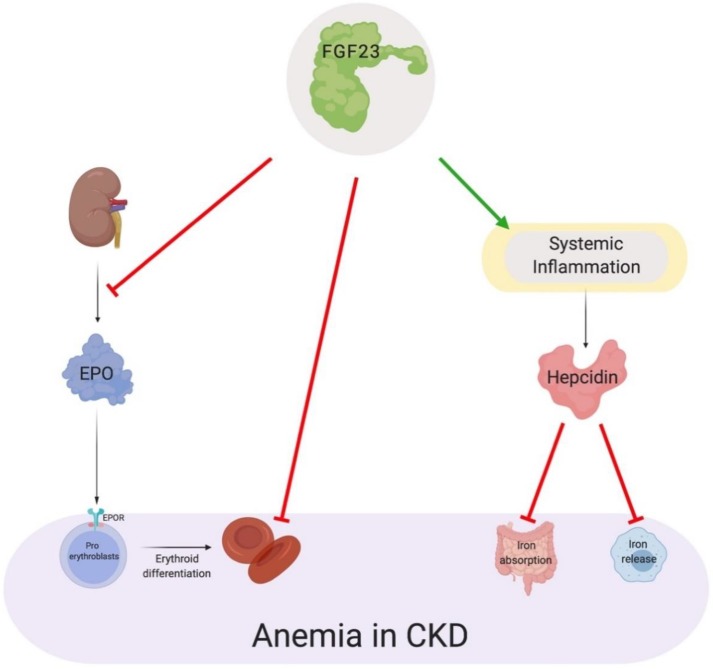
The pleotropic actions of FGF23 in anemia. As a direct consequence, FGF23 reduces the secretion of EPO from the kidney, thereby decreasing the differentiation of erythroid progenitors, such as pro-erythroblasts, to mature erythrocytes. In addition, FGF23 directly reduces the fraction of erythrocytes in the G2/M phase of their cell cycle and enhances erythrocyte apoptosis. As an indirect consequence, FGF23 enhances the inflammatory milieu, which in turn, promotes hepcidin excess and leads to the restriction of iron for erythropoiesis. EPO, erythropoietin; EPOR, erythropoietin receptor.

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
