# Peer review of "The Role of Fibroblast Growth Factor 23 in Inflammation and Anemia"

_ijms, 2019, doi:10.3390/ijms20174195_

Round 1

Reviewer 1 Report

The review article is interesting and very comprehensive. The graphical part is also attractive. However, more recently, a similar review has been published (FEBS Lett. 2019 Jun 14. doi: 10.1002/1873-3468.13494. Regulation of fibroblast growth factor 23 (FGF23) in health and disease).

Author Response

We would like to thank Reviewer #1 for the interest in our manuscript. We agree that other review articles with a similar focus have been already published, including the recent one in FEBS Letters. However, we think that our manuscript goes further into depth discussing the general role and effect of inflammation and iron metabolism in chronic kidney disease (CKD). In addition, we discuss the associations between inflammation, anemia and FGF23 in greater detail and how these three factors exhibit multi-directional relationships in the context of CKD.

Reviewer 2 Report

Brian Czaya and Christian Faul gave a very through and impressive review on “FGF23 in inflammation and anemia”. This includes evolutional point of views, molecular mechanisms, animal models and clinical overviews of FGF23 with focuses on phosphate metabolism, inflammation and anemia under CKD condition.

I only have a few suggestions.

It will help readers if more schematic graphs will be provided, particularly in the session of how FGF23 intact, N-term and C-term are processed (session 2, page 2 and 3). How the protease is regulated. Again, it will be great if there is a graph shows the canonical and non-canonical signaling of FGF23 (session 3, page 3 and page4). Minor points as following. Some terms were not defines (or errors) in the text, such as: FGFR1c (P4L141), Rap1 (P5L180), KL1 (P5L191), O2- (- was missing, P11L411), O2-, (P11L416), duplicate text (P27L1087-P27L1090).

Author Response

Reviewer: It will help readers if more schematic graphs will be provided, particularly in the session of how FGF23 intact, N-term and C-term are processed (session 2, page 2 and 3). How the protease is regulated. Again, it will be great if there is a graph shows the canonical and non-canonical signaling of FGF23 (session 3, page 3 and page4).

Response: We agree with Reviewer #2. We have generated two new figures showing the processing of FGF23 and the canonical versus non-canonical FGF23-mediated signaling. We have added the two figures to the revised manuscript.

Reviewer: Minor points as following. Some terms were not defined (or errors) in the text, such as: FGFR1c (P4L141), Rap1 (P5L180), KL1 (P5L191), O2(- was missing, P11L411), O2-, (P11L416), duplicate text (P27L1087-P27L1090). 

Response: All of these points have been corrected in the revised manuscript.